# FOXO3 Depletion as a Marker of Compression-Induced Apoptosis in the Ligature Mark: An Immunohistochemical Study

**DOI:** 10.3390/ijms24021396

**Published:** 2023-01-11

**Authors:** Aniello Maiese, Alice Chiara Manetti, Paola Santoro, Fabio Del Duca, Alessandra De Matteis, Emanuela Turillazzi, Paola Frati, Vittorio Fineschi

**Affiliations:** 1Department of Surgical Pathology, Medical, Molecular and Critical Area, Institute of Legal Medicine, University of Pisa, 56126 Pisa, PI, Italy; 2Department of Anatomical, Histological, Forensic and Orthopaedic Sciences, Sapienza University of Rome, Viale Regina Elena 336, 00161 Rome, RM, Italy

**Keywords:** immunohistochemistry, FOXO3, vitality, suicide, hanging

## Abstract

One of the most challenging issues in forensic pathology is lesion vitality demonstration, particularly in cases of hanging. Over the past few years, immunohistochemistry has been applied to this field with promising results. In particular, protein and transcription factors involved in the apoptotic process have been studied as vitality markers for the ligature mark. This study represents an implementation of our previous studies on ligature mark vitality demonstration. In this study, we evaluated the FOXO3 expression in post-mortem cervical skin samples through an immunohistochemical analysis. To evaluate FOXO3 expression, anti-FOXO3 antibodies (GTX100277) were used. The study group comprised 21 cases, 8 women and 13 men, whereas the control group consisted of 13 cases of subjects who died due to other causes. Decomposition and no clear circumstantial data were exclusion criteria. We found that FOXO3 is decreased in hanging cases compared with normal skin in other causes of death (*p*-value < 0.05). No differences were seen concerning the type of hanging material (hard or soft), type of hanging (complete or incomplete), and position of the knot. Our results suggest that FOXO3 depletion could be a valid immunohistochemical marker of ligature mark vitality.

## 1. Introduction

One of the most challenging issues in forensic pathology is lesion vitality demonstration, particularly in cases of hanging [1,2]. Some authors based the distinction of an ante-mortem or post-mortem lesion on the presence of inflammatory reaction and healing processes [3]. However, they are not always present, especially when the death occurred in a very short period of time [4]. The extravasation of blood within the tissues is one of the main histological findings in the hanging mark skin sample; however, it has been shown that it is possible to find red blood cells even in post-mortem injuries [5,6].

In recent years, immunohistochemistry (IHC) has been used to evaluate the protein expression in wounded tissues. The study conducted by Turillazzi et al. demonstrated that some cytokines and inflammatory cells expression is enhanced in vital skin lesions [2]. Ishida et al. studied 56 cases of asphyxia by compression of the neck (hanging and strangulation), proving that Aquaporin 3 (AQP3) was significantly expressed in the epidermis of vital ligature marks [7]. Ye et al. demonstrated a differential expression of IL-6 and IL-20 in wounded skin [8]. Another interesting work provided evidence of the enhanced expression of C-X-C motif chemokine ligand 1 (CXCL1) and C-X-C motif chemokine receptor 2 (CXCR2) in wounded skin, demonstrating that they may be used as vitality markers in forensic pathology. The evaluation of protein expression through the study of mRNA levels is very promising in this field [9].

It has already been demonstrated that the external pressure provoked by the ligature in hanging induces skin cell apoptosis due to ischemia [10,11]. Based on this evidence, Maiese et al. showed that the FLICE-inhibitory protein (c-FLIP), which has anti-apoptotic functions, is decreased in the ligature marks [12]. Recently, some researchers identified the cyclic AMP-dependent transcription factor (ATF3) as a vitality marker in skin contusion through an evaluation of its mRNA levels [13].

Also called programmed cell death, apoptosis is a controlled process that regulates the life cycle, human embryo development, and cell stress response. The balance between the expression of antiapoptotic and proapoptotic genes regulates cell homeostasis.

The Forkhead box O (FOXO) is a family of transcriptional factors involved in multiple cellular pathways [14,15]. There are four mammalian FOXO members that share a high protein homology [16]. Among these, FOXO3 induces apoptosis, regulating the transcription of pro-apoptotic or anti-apoptotic genes [17,18]. It is inactivated by phosphorylation, which induces its confinement into the cytoplasm, while it needs to be dephosphorylated and translocate into the nucleus to activate its target genes [19,20]. As a stress-induced response, various protein kinases, such as AKT, target FOXO3, which, in turn, controls various protein expression, such as the cellular FLICE (FADD-like IL-1β-converting enzyme)-inhibitory protein (c-FLIP) [21,22]. c-FLIP is expressed in several tissues and appears to have an anti-apoptotic role [23]. When a tissue is injured, crosstalk between c-FLIP and FOXO3 starts. It has been demonstrated that FOXO3 down-regulates c-FLIP. In cancer tissues, FOXO3 is phosphorylated and inactive, so it is unable to down-regulate c-FLIP and other anti-apoptotic genes; therefore, apoptosis is inhibited [24,25,26].

The aim of this research is to provide relevant vital signs of ligature marks, focusing on immunohistochemically new marker detectable vital reactions to provide a differentiation between ligature mark features and surrounding non-injured skin.

## 2. Results

The histologic examination of the ligature mark skin samples showed a flattening of the epidermal layers and the formation, in a few sections, of intra-epidermal liquid-filled vesicles. No foci of hemorrhagic infiltration were found in the epidermal layer. In a minority of sections, leukocytes within the dermal tissue and Zenker’s necrosis of the muscular layer were present. Dermal vessel plethora and connective tissue metachromasia were observed in skin samples obtained from post-mortem suspension cases.

Concerning the IHC analysis, in all 21 cases of hanging, the ligature mark skin samples, obtained at the point of greater compression (“full of the loop”), showed a depletion in FOXO3 in the epidermal layers in correspondence with the epidermal flattening (average value of intensity −2.81, *p*-value < 0.05), as shown in Figure 1A,B. In addition, the adjacent non-injured epidermis was positive to FOXO3a (value of intensity 0), which was localized both in the cytoplasm and nucleus (Figure 1C).

Table 1 shows the results of the semi-quantitative analysis of the immunohistochemical reaction in the injured skin section of the 21 cases of hanging. No qualitative or statistical differences were seen concerning the material used for hanging (hard or soft), the position of the knot, or the type of hanging (complete or incomplete).

All control skin specimens, obtained from the neck of subjects who died of other causes, showed FOXO3 positivity both in the cytoplasm and in the nucleus of epidermal cells (Figure 1D, intensity score 0). Skin samples obtained from the neck of subjects suspended after the death showed the same pattern of FOXO3 positivity as well. Statistical analysis via the Student test showed a statistically significant FOXO3 depletion for hanging cases compared to post-mortem injuries and uninjured skin specimens (*p* < 0.05).

## 3. Discussion

The results presented in this study demonstrate that, in the ligature mark epidermis, there is a depletion in FOXO3 if compared to normal or post-mortem injured skin (*p* < 0.05). FOXO3 depletion was detected both in soft and hard tissue, indicating the same underling ischemia-induced mechanism. These results are in accordance with our previous work, conducted on the same study group, in which we demonstrated that, in the ligature mark, there is a hypo-expression of c-FLIP [12]. Skin compression causes ischemia of the epidermis [3]. This determines the inhibition of protein kinases, such as AKT, that phosphorylate FOXO3. [27] Accordingly, FOXO3, which is confined in the cytoplasm, ubiquitinated, and degraded by the proteasome when phosphorylated, is no more phosphorylated and thus is able to enter the nucleus. When FOXO3 enters the nucleus, it inhibits the transcription of c-FLIP (Figure 2) and promotes the expression of pro-apoptotic genes, such as Bim and Puma [22,28,29,30,31]. Therefore, FOXO3 has a pro-apoptotic activity in the ligature mark epidermis.

We already demonstrated in our previous work that c-FLIP is hypo-expressed in the ligature mark epidermis [12].

Therefore, the question is: if c-FLIP and FOXO3 have opposite roles in the apoptotic process (inhibiting and promoting apoptosis, respectively), why are both of them decreased/absent in the ligature mark epidermis? Once FOXO3 has promoted the transcription of its target gene, it is phosphorylated at the 14-3-3 proteins and extruded from the nucleus. Herein, it is ubiquitinated and degraded [32]. We hypothesize that the hyperactivity of FOXO3 in the ischemic epidermis of the ligature mark causes its depletion and, as a result, the IHC reaction is negative. Certainly, the interaction between FOXO3, c-FLIP, and the other proteins involved in the apoptotic cascade needs to be further investigated.

The greater limitation of the current study is the small sample size. Only 21 cases of hanging were suitable for the research. In further studies, we aim to enlarge our population to validate our findings using IHC and immunofluorescence methods. This study surely demonstrates the promising potential of the apoptosis molecules as markers of vitality in the ligature mark skin. This research group aims to validate an antibody toolkit to be used to confirm the differential diagnosis between ante-mortem and post-mortem lesions. Indeed, in forensic practices, it is often difficult to demonstrate the vitality of the ligature mark. In other words, it is not always possible to state if the skin compression of the neck has been produced ante-mortem or post-mortem. Former studies [33] demonstrated that various markers could be used to differentiate vital skin lesions from post-mortem ones, such as AQP3, MHC-II, avidin, CD1a, cathepsin D, and P-selectin [7,34,35]. Regarding our research team, firstly, we highlighted the reliability of tryptase, IL-15, and CD15 in the determination of ligature marks’ vitality [2]. Then, we evaluated the use of Troponin I—fast skeletal muscle (TNNI2), which was greatly expressed in ligature mark skin samples [36]. With the improvements in molecular biology, we also studied microRNAs (miRNAs) expressed in skin sampled from hanging ligature marks [10]. We found an overexpression of miRNA with anti-inflammatory activity (miR214a-3p, miR128-3p, miR130a-3p, and miR92a-3p) and miRNA involved in the inflammatory response (miR125a-5p and miR125b-5p). The use of miRNAs in forensic pathology and wound vitality demonstration seems to be promising and further research is focalizing on this topic [11,37].

## 4. Materials and Methods

### 4.1. Study Group Selection and Samples Collection

The autopsy databases of the Legal Medicine and Forensic Institutes of the “Sapienza” University of Roma and University of Pisa were retrospectively reviewed. Analyzing the autopsy reports and the information gathered from the police investigation, 21 cases of suicidal hanging deaths were selected. Cases of decomposed body (even initial signs of putrefaction) or those with unclear circumstantial data about the manner of death were excluded. The resulting study group was composed of 8 women and 13 men, with a mean age of 52.2 years. Among these 21 cases, in 11 cases, broad, soft, and yielding materials were used as a hanging tool (e.g., sheets), and in the remaining 10 cases, the ligature material was hard (e.g., a rope). The knot was situated over the back of the neck (occipital area) in 7 cases, on the right side of the neck (right mastoid) in 10 cases, and on the left side of the neck (left mastoid) in 4 cases. Hanging was complete in 15 cases and incomplete in 6 cases.

As a control group, 13 other cases were chosen (6 women, 7 men, mean age of 47.3 years). Samples of apparently healthy skin were collected at later time. Decomposed bodies or those with initial signs of putrefaction were excluded. The causes of death were drug overdoses in 5 cases, car accidents in 3 cases, and sudden cardiac deaths in 5 cases. In 3 out of 5 cases of drug overdoses, post-mortem suspension of the body occurred.

Each body was autopsied within 36 h after the death. During the autopsy, skin samples were collected from each case. In all cases of hanging (post-mortem and ante-mortem), sections of skin were removed from the neck at the site of the greater depth of the marks, whereas, in control cases, the skin samples were taken from the anterior face of the neck.

This study represents an implementation of our previous studies, and the study group and skin samples are the same as described in our previous work on apoptosis molecules expression in the ligature mark [12].

The processing of the data reported in this paper is covered by the general authorization to process personal data for scientific research purposes granted by the Italian Data Protection Authority (1 March 2012 as published in Italy’s Official Journal no. 72 dated 26 March 2012) since the data do not entail any significant personalized impact on data subjects. Our study did not involve the application of experimental protocols; therefore, it did not require approval by an institutional and/or licensing committee. The bodies included in this study were autopsied by order of the Italian Judicial Authority. In all cases, local prosecutors opened an investigation, ordering an autopsy to be performed to clarify the exact cause of death. Therefore, according to the Italian law, no ethical approval was needed. However, the present study was conducted with respect of the deceased involved and any data were anonymized to guarantee the privacy of each subject.

### 4.2. Histological and Immunohistochemical Analysis

A routine microscopic histopathological study was performed using hematoxylin-eosin (H&E) staining. In addition, an IHC investigation of skin samples was performed, as previously published [2,10,12].

Skin fragments measuring 0.5 cm × 0.8 cm were collected. Samples, 8 cm^2^, from each case were fixed in 10% buffered formalin and then washed with phosphate-buffered saline (PBS), and subsequent dehydration was carried out using a graded alcohol series. Time in formalin was variable, from a minimum of 7 h to a maximum of 72 h. After dehydration, samples were cleared in xylene and embedded in paraffin. Sections were cut at 4 μm, mounted on slides, and covered with 3-amminopropyltriethoxysilane (Fluka, Buchs, Switzerland).

To evaluate FOXO3 expression, anti-FOXO3 antibodies (GTX100277) were utilized. They recognize both the phosphorylated and dephosphorylated forms of FOXO3. Antigen retrieval was carried out using EDTA buffer in a pressure steamer at 100 °C for 90 min. Antigen retrieval was automated with Dako PT link [buffer pH 9]. Slides were stained on an automated immunostainer (Dako Cytomation, Glostrup, Denmark) using a polyclonal anti-FOXO3 antibody (GeneTex cat. No. GTX100277 FOXO3A Ab (C3), C-term (knockout (KO) validated). (Tryptase: 5 min Proteolytic Enzyme (Dako, Copenhagen, Denmark), 20 °C 120 min, 20 °C 1:1000. CD 15: (DAKO, Copenhagen, Denmark) boiling in 0.25 mM EDTA buffer; 120 min, 20 °C 1:50.)

Before staining the study group’s samples, anti-FOXO3a antibodies (GTX100277) were tested on breast cancer samples. As other authors already demonstrated, FOXO3 is positive and could have a nuclear and cytoplasmatic localization in such tissues [38]. In Figure 3, the breast cancer positive control is shown.

### 4.3. Quantitative Analysis

As described in our previous study, each immunohistochemical slide underwent a quantitative analysis [39]. For quantitative analysis, in each immunohistochemical section, 20 observations were made in different fields/slides at 100-fold magnification. The samples were also examined under a confocal microscope and a three-dimensional reconstruction was performed (True Confocal Scanner, Leica TCS SPE, Cambridge, UK).

The staining intensity was evaluated using a semi-quantitative scoring scale. A semi-quantitative blind evaluation of the IHC findings by two different investigators (AM and VF) was performed. All measurements were carried out at the same magnification of image (×10) and the gradation of the immunohistochemical reaction was used with a scale from 0 to −3, as shown in Table 2. The grade was based on the maximum depletion of FOXO3 noted. The evaluations were carried out separately for each sample using a double-blind method. In cases of divergent scoring, a third observer (ET) decided the final score.

### 4.4. Statistical Analysis

Semi-quantitative evaluation of the IHC findings and gradation of the immunohistochemical reaction were described with an ordinal scale. The median values were then reported. Analysis of variance for the non-parametric data was performed using Kruskal–Wallis test. When differences were found to be significant, analysis between the unmatched groups was elucidated with a Dunn’s multiple comparison post hoc test. The significance level was set to 5% (SPSS ver. 16.01 for Windows—SPSS Inc., Chicago, IL, USA).

## 5. Conclusions

In this study, we demonstrated that FOXO3 depletion could be a valid immunohistochemical marker of ligature mark vitality. Differential diagnosis between ante-mortem and post-mortem injuries is crucial in forensic pathology. The current literature is determined to find molecular markers that are reliable in establishing the vitality of the ligature mark. Scientific research in forensic medicine should identify a molecular marker with features such as objectivity of evidence, a high reproducibility, being easy to find, and scientifically validated so that it can be used as evidence in judicial trials. Circumstantial data, crime scene investigation, and forensic study of the deceased are the first steps used to demonstrate the vitality of a lesion. However, a histopathological analysis of the ligature marks is not always enough to differentiate if it has been produced ante-mortem or post-mortem. Following our previous studies, we suggest using a panel of markers, such as tryptase, IL-15, troponin I, c-FLIP, FOXO3, and miRNAs, in cases of hanging to confirm or deny the vitality of the compression lesion [40,41]. Further studies are certainly needed to evaluate which combination of markers should be used or if there are further useful markers that have not been investigated yet.

## Figures and Tables

**Figure 1 ijms-24-01396-f001:**
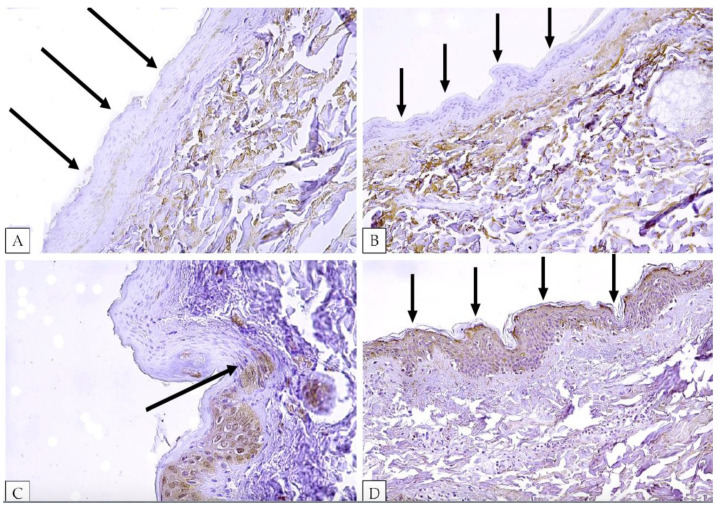
FOXO3 expression in our skin samples. (**A**,**B**), FOXO3 is negative (black arrows) in the epidermal layer of skin samples obtained from the “full of the loop” part of the ligature mark. (**C**), FOXO3 is negative in the compressed epidermis, but it is positive (black arrow to signify the axe blow of differentiation) both in the cytoplasm and nucleus in the adjacent non-injured, skin. (**D**), normal skin epidermis shows positivity (value of intensity 0 according to our grading system) for FOXO3 (black arrows).

**Figure 2 ijms-24-01396-f002:**
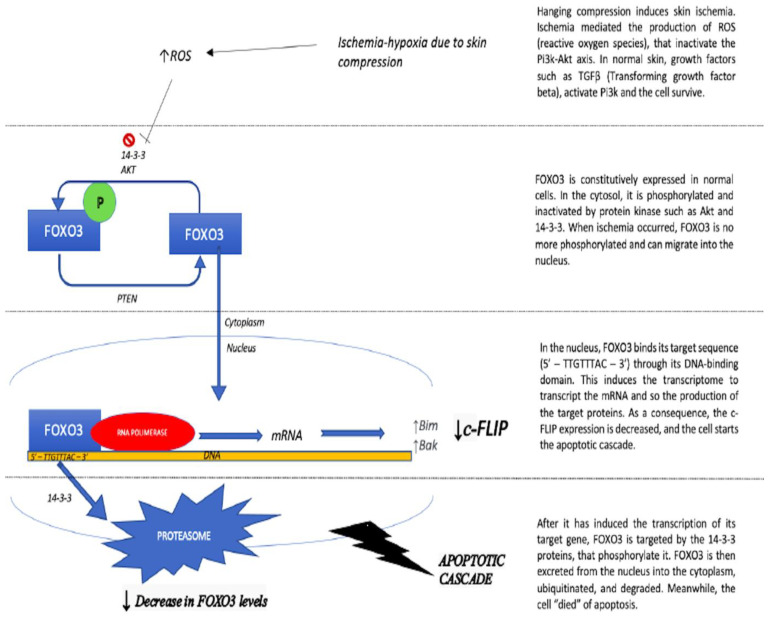
Mechanism of action of FOXO3 in its molecular steps.

**Figure 3 ijms-24-01396-f003:**
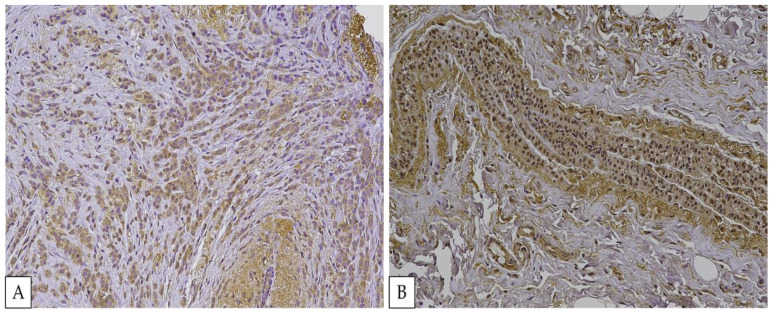
FOXO3 positivity in breast cancer. (**A**) FOXO3 is predominantly localized in the cytoplasm. (**B**) FOXO3 is localized both in the cytoplasm and nucleus.

**Table 1 ijms-24-01396-t001:** Semi-quantitative analysis of the staining intensity concerning the circumstantial data (type of hanging material, type of hanging, position of the knot). The average intensity value is −2.81.

Case Number	Sex	Staining Intensity	Hanging Material	Type of Hanging	Knot Position
1	M	−3	Soft	Complete	Occipital
2	F	−3	Soft	Incomplete	Occipital
3	M	−2	Soft	Complete	Right/Left side of the neck
4	M	−3	Hard	Incomplete	Right/Left side of the neck
5	M	−3	Hard	Complete	Occipital
6	M	−3	Soft	Complete	Occipital
7	F	−3	Hard	Incomplete	Occipital
8	F	−2	Soft	Complete	Right/Left side of the neck
9	F	−3	Soft	Complete	Right/Left side of the neck
10	M	−3	Soft	Complete	Right/Left side of the neck
11	M	−2	Soft	Complete	Right/Left side of the neck
12	F	−3	Soft	Incomplete	Right/Left side of the neck
13	M	−3	Hard	Complete	Right/Left side of the neck
14	M	−3	Hard	Complete	Right/Left side of the neck
15	M	−3	Hard	Incomplete	Right/Left side of the neck
16	F	−2	Soft	Complete	Occipital
17	M	−3	Hard	Complete	Occipital
18	M	−3	Hard	Incomplete	Right/Left side of the neck
19	M	−3	Soft	Complete	Right/Left side of the neck
20	F	−3	Hard	Complete	Right/Left side of the neck
21	F	−3	Hard	Complete	Right/Left side of the neck

**Table 2 ijms-24-01396-t002:** The amount and extent of marker depletion were scored for each section from 0 to −3. The interpretation of our scoring system is shown in this table.

Grade	Interpretation
0	No loss of staining (positivity)
−1	Minimal decrease in staining compared to normally stained tissue
−2	A clear decrease in staining with some positivity (brown color) remaining
−3	No positive staining

## Data Availability

The data presented in this study are available in Table 1.

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
