# Peer review of "FOXO3 Depletion as a Marker of Compression-Induced Apoptosis in the Ligature Mark: An Immunohistochemical Study"

_ijms, 2023, doi:10.3390/ijms24021396_

Round 1

Reviewer 1 Report

I read the proposed manuscript with great interest. The article is well-written and fits into a very relevant research area 

However, there are a few aspects that the authors should clarify:

- It is not clear what type of study the authors conducted. They claim to have done a retrospective study of people who had died by hanging and from whom they had taken a skin fragment from the skin furrow at autopsy. That's plausible because it's always appropriate to take and retain those kinds of samples. But what about the control samples? Were they also already collected or were they collected prospectively after the study began?

- One important element that the authors omit is how old the skin samples they analyzed from the subjects who died from hanging were. The skin samples were obviously preserved in formalin, but it is known that long residence time in formalin can lead to the masking of antigens. This may affect subsequent immunohistochemical analyses if appropriate unmasking techniques are not used. This aspect needs to be clarified and possibly considered in discussions 

- Of all the findings that are expressions of vitality in a hanging furrow, the authors fail to mention the most important one, which is hemorrhagic infiltration (detectable in immunohistochemistry with anti-Glycophorin A antibody). In the results, hemorrhagic infiltrates are not described, which is an anomaly. Moreover, given the experimental study of a molecular marker as an index of vitality, it would have been highly appropriate if the authors had also accompanied the assessment of hemorrhagic infiltrate to strengthen the evidence base. Because staining with H&E has already been performed, an assessment of the presence of the hemorrhagic infiltrate (even without the use of immunohistochemistry) would be a great improvement for the robustness of the reported evidence.

- The authors state that the small number of samples (21) is a limitation of the study. Why then did the authors limit themselves to only 21 samples? The study involved two important Italian centers (Rome and Pisa), and the number of suicides by hanging is unfortunately very high.

- In Material and Methods, there is a sentence without clear meaning 'In three out of five cases of drug overdoses, post-mortem suspension of the body occurred' If these were suicides, how could postmortem hanging occur? Furthermore, these cases are not highlighted in Table 2. Clarification would be needed here.

- The authors could report the dimensions of the specimens in centimeters and not just in terms of surface area.

- The distinction between - 1 (minimal decrease) and -2 (clear decrease) is not based on objective criteria. Why did the authors not use a precise numerical value, e.g. a percentage decrease? This makes it difficult to repeat the work because an objective criterion is missing. It would also be appropriate to specify the qualifications of the observers. Given the sensitivity of the observations, experienced pathologists (not forensic, but with diagnostic skills) should have been involved.

- the authors noted no difference between skin furrows caused by hard or soft material. This is a curious finding because the different instrument exerts a different force on the skin and thus intuitively on the underlying cells. Therefore, the authors should discuss this finding in the discussions, which is missing in the current version. Did the authors also try to consider the weight of the victims as a potential influencing factor?

Overall, the results are interesting, some improvements would be necessary, otherwise, the soundness of the methodological approach of this study could be questioned.

Author Response

Thank you very much for your comments. We highly appreciate the detailed valuable comments on our manuscript of “FOXO3 depletion as a marker of compression-induced apoptosis in the ligature mark: an immunohistochemical study”.
The suggestions were helpful for us and we incorporate them in the revised paper. We tried our best to revise it and we hope these efforts will be worked. A point-by-point description of the revisions made follows:

However, there are a few aspects that the authors should clarify:
- It is not clear what type of study the authors conducted. They claim to have done a retrospective study of people who had died by hanging and from whom they had taken a skin fragment from the skin furrow at autopsy. That's plausible because it's always appropriate to take and retain those kinds of samples. But what about the control samples? Were they also already collected or were they collected prospectively after the study began?
•    Response: we clarified that this is a retrospective study on all cases of suicidal hangings. Skin samples of hangings were collected and stored for legal purposes. Control samples come from the same database, from which healthy skin samples were taken [see cases in lines 81-84]. Obviously, control samples were collected at a later time. In the text we added the specification “As a control group, other 13 cases were chosen (six women, seven men, mean age of 47.3 years) and collected samples of apparently healthy skin at later time.”

- One important element that the authors omit is how old the skin samples they analyzed from the subjects who died from hanging were. The skin samples were obviously preserved in formalin, but it is known that long residence time in formalin can lead to the masking of antigens. This may affect subsequent immunohistochemical analyses if appropriate unmasking techniques are not used. This aspect needs to be clarified and possibly considered in discussions. 
•    Response: We understand your concern and we have better explained this point. Thank to this comment, we noticed that we described when the samples were collected, but not how long they were in formalin and how long it has been since they were embedded in paraffins. 
We modify line 115: “Time in formalin was variable, from minimun 7 hours, to a maximum of 72 hours.” and line 123: “Antigen retrieval is automated with Dako PT link [buffer pH 9]”.

- Of all the findings that are expressions of vitality in a hanging furrow, the authors fail to mention the most important one, which is hemorrhagic infiltration (detectable in immunohistochemistry with anti-Glycophorin A antibody). In the results, hemorrhagic infiltrates are not described, which is an anomaly. Moreover, given the experimental study of a molecular marker as an index of vitality, it would have been highly appropriate if the authors had also accompanied the assessment of hemorrhagic infiltrate to strengthen the evidence base. Because staining with H&E has already been performed, an assessment of the presence of the hemorrhagic infiltrate (even without the use of immunohistochemistry) would be a great improvement for the robustness of the reported evidence.
•    Response: thanks for the suggestions. We read again histological specimens. Hemorrhagic infiltration was not detected in all samples. Red blood cells in soft tissue near the lesioned skin are indicative of a trauma, so they could help to confirm the diagnosis, but are not pathognomonic. Other signs of vitality and circumstantial data made it possible to reach the diagnosis and confirm it. The detection of small hemorrhagic infiltrates in some samples was useful but not necessary. What is observed under the microscope is reflected with the data of specific literature on the ligature furrow. Moreover, in recent literature [https://doi.org/10.1177/00258024211023246], Gentile G. Zoja R. et Al have shown that is rare to find foci of hemorrhagic infiltration in the epidermal layer. Our study focuses solely on the epidermis of the hanging furrow, so we did not consider it useful to mark it.
However, we decided to follow your advice and to add in the text: Line 163 “No foci of hemorrhagic infiltration were found in the epidermal layer.”
- The authors state that the small number of samples (21) is a limitation of the study. Why then did the authors limit themselves to only 21 samples? The study involved two important Italian centers (Rome and Pisa), and the number of suicides by hanging is unfortunately very high.
Response: Although Rome and Pisa are two important Italian centers, most suicides by hanging are not entrusted to the coroner on behalf of the judicial authority. In these numerous cases, an autopsy is not performed in our morgues, as there is no crime. Thus, the number of autopsies in hangings is very low.For what concern number of cases, we underline that we must follow the same protocol of previously study “Maiese, A.; De Matteis, A.; Bolino, G.; Turillazzi, E.; Frati, P.; Fineschi, V. Hypo-Expression of Flice-Inhibitory Protein and Activation of the Caspase-8 Apoptotic Pathways in the Death-Inducing Signaling Complex Due to Ischemia Induced by the Compression of the Asphyxiogenic Tool on the Skin in Hanging Cases. Diagnostics. 2020, 10, 938. doi: 10.3390/diagnostics10110938.”. In order to make comparable the two studies we have examined same samples. In the study design we had to choose the same samples present in the paper, to compare the previous results. In the future we plan to expand the study to larger numbers of samples examined.

- In Material and Methods, there is a sentence without clear meaning 'In three out of five cases of drug overdoses, post-mortem suspension of the body occurred' If these were suicides, how could postmortem hanging occur? Furthermore, these cases are not highlighted in Table 2. Clarification would be needed here.
•    Response: Thanks for the suggestion; drug overdose occurred accidentally, not as a suicide mode. In three cases, somebody try to stage a suicide with postmortem suspension. In other words, we collected samples of five drug overdose deaths, of which three were simulated hangings and two were found dead without any special features.
For a better understanding, we have specified that we have taken the skin of the furrow even in cases of simulated hanging, while in control cases we have taken only healthy skin. See line 98: “In all cases of hanging (post-mortem and ante-mortem)”. 
- The authors could report the dimensions of the specimens in centimeters and not just in terms of surface area.
•    Response: we add skin fragment dimension in centimeters in line 113 “Skin fragments measuring cm 0.5 x cm 0.8 were collected.”
- The distinction between - 1 (minimal decrease) and -2 (clear decrease) is not based on objective criteria. Why did the authors not use a precise numerical value, e.g. a percentage decrease? This makes it difficult to repeat the work because an objective criterion is missing. It would also be appropriate to specify the qualifications of the observers. Given the sensitivity of the observations, experienced pathologists (not forensic, but with diagnostic skills) should have been involved.
•    Response: As highlighted, to give more resonance to the study we decided to conduct it with the same samples and methods as the previous one. In this way it was possible to compare data. immunohistochemical slides were read in double-blind by two forensic pathologists [VF and AM] with extensive diagnostic experience. Disagreements were resolved by a third pathologist.
- the authors noted no difference between skin furrows caused by hard or soft material. This is a curious finding because the different instrument exerts a different force on the skin and thus intuitively on the underlying cells. Therefore, the authors should discuss this finding in the discussions, which is missing in the current version. Did the authors also try to consider the weight of the victims as a potential influencing factor?
•    Response: The ligature mark differs according to the hard or soft materials that produces it. Not always microscopic features are associated with external examination of the furrow. A "hard" and "soft" sulcus have different macroscopic characteristics, but on histological examination they occur in a similar way [detachment of the corneous layer, ectasia of the dermal vessels, flattening of the nuclei...]. This explain why there are no differences in FOXO3 expression. Line 196: “FOXO3 depletion was detected both in soft and hard furrow, indicating the same underling ischemia-induced mechanism.”

Reviewer 2 Report

One of the most challenging issues in Forensic Pathology is lesion vitality demonstration, particularly in cases of hanging. In this study, the authors evaluated the FOXO3 expression in post-mortem cervical skin samples through an immunohistochemical analysis. Their results suggest that FOXO3 depletion could be a valid immunohistochemical marker of ligature mark vitality. The manuscript is clearly written, the conclusions are well supported by the data. However, the authors might need to revise the manuscript in order to increase its clarity and readability, considering the following points.

1. The Introduction section may be enhanced through revisions that offer a more comprehensive review of the relevant background and highlight the potential challenges in evaluating wound vitality.

2. It is important for authors to address ethical considerations in their research, particularly when it involves human subjects. In this case, as the study involves forensic autopsy cases, it is necessary for the authors to provide information about the institutional ethics clearance obtained for the study. This is particularly important as it demonstrates that the research was conducted in accordance with ethical standards and guidelines. The inclusion of an ethics statement in the manuscript will also help to ensure the credibility and transparency of the research.

3. It is suggested that the authors consider incorporating semi-quantitative methods (WB) for analyzing FOXO3 and examining changes in FOXO3 mRNA, in addition to utilizing immunohistochemical staining, if feasible. Such techniques may provide additional insights for the practical application of forensic pathology.

4. It is important for authors to thoroughly review and cite relevant literature in their research. While the cited literature in the manuscript is appropriate, it appears that some fundamental and important papers have been omitted, such as those with PMID numbers 31546162, 30059828, 29607464, and 30483664. It is recommended that the authors carefully review their literature review and consider adding these additional references to provide a more comprehensive overview of the current state of knowledge in the field.

5. It is important for authors to include negative controls in their experiments to ensure the validity and reliability of the results. In this case, it appears that Fig. 2, which represents the immunostaining results, lacks negative controls. It is recommended that the authors include negative controls in Fig. 2 to provide a more complete and accurate depiction of the experimental results. This will help to ensure the credibility and transparency of the research.

6. It is suggested that the authors consider adding the assessment of apoptosis-related indicators to their experimental design in order to refine the study and validate the hypothesis proposed.

Author Response

Dear reviewer, thank you for your comments. We modified the manuscript according to your considerations. Now we think it very improves. 
1) In the introduction, we added some sentences, implementing the review of the previous papers about this topic (lines 32-48). 
2) We perfectly agree with you about the importance of ethics in such kind of studies. However, the bodies included in this study were autopsied by order of the Italian Judicial Authority. Therefore, according to the Italian law, no ethical approval was needed. Anyway, the present study has been conducted with respect of the deceases involved and any data has been anonymized to guarantee the privacy of each subject. We added a few lines in Materials and Methods to clarify this point (lines 101-111).
3) thank you for your advice. Our research team would like to implement this research, using additional methodology and techniques. We hope we will be able to show this implementation in the next studies.
4) Thank you, we added the references suggested. They were very appropriate.
5) We agree with the fact that negative controls are very important in any experimental study. In figure 1, we showed the positive control, the expression of FOXO3 in a mammalian human cancer. We used it as a positive control because we found in the literature that FOXO3 is expressed in such kind of tumors. In figure 2, we showed the comparison between normal (not injured) skin (Fig. 2D) and ligature mark samples (Fig.2 A-C). The studies about FOXO3 are kind of a novelty, especially in this field. In the literature, we could not find any indication about a specific tissue in which FOXO3 is depleted. Indeed, in the discussion we provide our hypothesis why we found such differential expression of FOXO3. In conclusion, we did not test any further tissue (as negative control) because there is no indication in the previous literature on which kind of tissue is FOXO3 negative. We hope this explanation is clear. In further study, with increasing evidence on FOXO3 expression, we hope we will implement this point. 
6) Thank you for this observation. In further studies, we will investigate other apoptosis-related indicators, as our project is to collect evidence about various apoptosis-related markers. Indeed, we have already conducted an analogous study using c-FLIP, an anti-apoptotic protein.

Round 2

Reviewer 1 Report

I thank the authors for introducing the requested clarifications. I have no further suggestions.

Reviewer 2 Report

The authors have thoroughly revised their manuscript, which is now suitable for publication.